# Conditional nonlinear optimal perturbations based on the particle swarm optimization and their applications to the predictability problems

Qin ZHENG [1], Zubin YANG [1], Jianxin SHA [2] and Jun YAN [1]

[1]Institute of Science, PLA University of Science and Technology, Nanjing, 211101, China

[2]Troop 94906, People's Liberation Army, Suzhou, 215157, China

*Correspondence to*:   Zubin YANG (qinzheng@mail.iap.ac.cn)

**Abstract.** In predictability problem research, the conditional nonlinear optimal perturbation (CNOP) describes the initial perturbation that satisfies the certain constraint condition and causes the largest prediction error at the prediction time. The CNOP has been successfully applied in estimation of the lower bound of maximum predictable time (LBMPT). Generally, CNOPs are calculated by a gradient descent algorithm based on the adjoint model, which is called ADJ-CNOP. This study, through the two-dimensional Ikeda model, investigates the impacts of the nonlinearity on ADJ-CNOP and the corresponding precision problems when using ADJ-CNOP to estimate the LBMPT. Our conclusions are that: (1) when the initial perturbation is large or the prediction time is long, the strong nonlinearity of the dynamical model on the prediction variable will lead to failure of the ADJ-CNOP method; (2) when the objective function has multiple extreme values, ADJ-CNOP has large probability to produce local CNOPs, hence make false estimation on the LBMPT. Furthermore, the particle swarm optimization (PSO) algorithm, one kind of intelligent algorithms, is introduced to solve this problem. The method using PSO to compute CNOP is called PSO-CNOP. The results of numerical experiments show that even with a large initial perturbation and long prediction time, or the objective function has multiple extreme values, PSO-CNOP can always obtain the global CNOP. Since the PSO algorithm is a heuristic search algorithm based on the population, it can overcome the impact of nonlinearity and the disturbance from multiple extremes of the objective function. Besides, to check the estimation accuracy of the LBMPT presented by PSO-CNOP and ADJ-CNOP, we partition the constraint domain of initial perturbations into sufficiently fine grid meshes, and take the LBMPT obtained by the filtering method as a benchmark. The result shows that the estimation presented by PSO-CNOP is closer to the true value than the one by ADJ-CNOP with the forecast time increasing.

## 1. Introduction

Weather and climate predictability problems are attractive and significant in atmospheric and oceanic sciences. The goal of studying predictability problems is to investigate and understand the reasons for and mechanisms of the prediction uncertainty. Early in 1975, Lorenz divided the predictability problem into two types based on the consideration that prediction uncertainties were mainly caused by the initial and model errors. The first type of the predictability problem is concerned with the uncertainty of the forecast results induced by initial errors, and the second type deal with the uncertainty caused by model errors. He further introduced the singular vector (SV) into the predictability problem study. The leading singular vector is the initial perturbation with the greatest linear growth, accordingly, can be used to estimate the evolution of initial errors in the tangent linear regime during the course of a forecast (Buizza and Palmer 1995; Lacarra and Talagrand 1988; Farrell 1990; Borges and Hartmann 1992; Kalnay 2003). Since the atmospheric and oceanic movement is a nonlinear physical process, SV based on linear theory cannot effectively express the procedure. This limits its applications to the predictability problems (Mu et al., 2003). To this end, Mu et al. (2003) presented the conditional nonlinear optimal perturbation (CNOP) method. CNOP refers to the initial perturbation that satisfies the certain constraint condition, and has the maximum nonlinear evolution at the prediction time. Therefore, it stands for the initial uncertainty which leads to the largest prediction error. CNOP has been widely applied to weather and climate predictability studies (Duan et al., 2004; Mu et al., 2007; Terwisscha and Dijkstra, 2008; Duan et al., 2009; Mu et al., 2009; Yu et al., 2012; Wang et al., 2012, 2013). Riviere et al. (2009) used an extension of the CNOP approach, i.e., the nonlinear singular vectors, to estimate the predictability of atmospheric moisture processes so that to reveal the effects of non-linear processes.

Based on actual demands, Mu et al. (2002) separated the predictability problem into three sub-problems, i.e., the problem of the LBMPT, the problem of the upper bound of maximum prediction error, and the problem of the lower bound of maximum allowable initial error and parameter error. Duan and Luo (2010) formulated these three problems into three constrained nonlinear optimization problems. Meanwhile, they used the CNOP method to find the solutions of the three sub-problems.

Capturing CNOPs is a kind of constraint optimization problem, and optimization algorithms commonly used in solving CNOPs are based on the gradient descent method, including the spectral projected gradient 2 (SPG2; Brigin et al., 2000), the sequential quadratic programming (SQP; Powell, 1982), limited memory BFGS (L-BFGS; Liu and Nocedal, 1989), etc. Among these algorithms, the gradient information is always provided by

backward integral of the corresponding adjoint model of the prediction model (Duan et al., 2004; Duan et al., 2008; Mu and Zhang, 2006; Mu et al., 2009; Jiang and Wang, 2010; Yu et al., 2012; Wang et al., 2012, 2013). But the optimal algorithms based on gradient information involve forward integral of the tangent model and backward integration of the adjoint model. It might cause the following two problems: (1) for the numerical prediction model with complex physical process, the validity of the tangent linear approximation cannot be guaranteed when the forecast period is long; (2) for actual prediction model, it is quite difficult and time consuming to develop the adjoint model. Recently, Zheng et al. (2012, 2014) attempted applying genetic algorithms (GAs) to capture CNOPs of the dynamical model containing discontinuous "on-off" switches. They concluded that GAs, with proper genetic operator configuration, can overcome the non-smooth influences and obtain the global CNOP with high probability. Thus, in non-smooth cases, using GAs to solve predictability problems is more effective than using the conventional optimization algorithm.

Particle Swarm Optimization (PSO) algorithm is an intelligent algorithm proposed by Kennedy and Eberhart (1995), which imitated the process of birds foraging. In PSO algorithm, each particle, as is a vector in solution space, represents a potential solution of the optimal problem. Compare to the gradient descent algorithm based on the adjoint model, the PSO algorithm has better effectiveness to search the global optimal solution for nonlinear and non-smooth optimal problems; the PSO do not require a gradient of the objective function. The PSO is similar to the GA in the sense that they are both population-based search approaches and that they both depend on information sharing among their population members to enhance their search processes using a combination of deterministic and probabilistic rules (Hassan et al., 2005). The PSO has the same effectiveness as the GA but with significantly better computational efficiency, it has memory and constructive cooperation between particles (Fang et al., 2009). This paper tries to use the PSO to calculate the CNOP of the forecast model, and obtains highly-precise estimation of the lower bound of maximum predictable time.

The Ikeda model was originally proposed by Kensuke Ikeda (1979) as a model describing light going across a nonlinear optical resonator. The Ikeda model has strong nonlinearity, and the two-dimensional difference scheme is its most common form. Li and Zheng (2016) investigated the stability of solutions of the Ikeda model, and tested the dependence of the solutions on the model parameter. In addition, they provided the solution description of various shapes corresponding to parameter values of different regions. This paper takes the two-dimensional Ikeda model as the prediction model to reveal how the nonlinearity impacts the precision when estimating the LBMPT based on the ADJ-CNOP method. A new method, PSO-CNOP, is presented to solve this problem.

This paper is organized as follows: section 2 is devoted to describe the three predictability sub-problems, the definition of CNOP and the two-dimensional Ikeda model. Section 3 provides the knowledge about the particle swarm optimization (PSO) algorithm. In section 4, the performances of ADJ-CNOP and PSO-CNOP are compared when solving CNOPs through numerical experiments. The impacts on the estimation precision of the lower bound of maximum predictable time are also demonstrated in this section. The conclusion and discussion are presented in section 5.

## 2. Related conceptions and the forecast model

The three predictability sub-problems, the definition of CNOP and the two-dimensional Ikeda model are briefly described as follows. And more detailed introductions can be referred to (Mu et al., 2002, 2003; Li and Zheng, 2016).

### 2.1 Three sub-problems of predictability problem

The three predictability sub-problems associated with the first kind of the predictability problem are introduced, and the forecast model is supposed to be perfect in the following.

Problem 1 The lower bound of maximum predictable time (LBMPT)

Assuming that there is an error in the initial condition (IC) of the forecast model, it will lead to a prediction error when integrating forward the model from the IC to predict the atmospheric or oceanic states in the future.

Let $u_0$ be the IC, $u_T^t$ the true state at time $T$, $\mathbf{M}_T$ the nonlinear propagator of the numerical forecast model from 0 to $T$ time, then under the assumption of the perfect model, $u_T^t = \mathbf{M}_T(u_0^t)$, where $u_0^t$ is the true state at the initial time.

With a given prediction precision $\varepsilon > 0$, the maximum predictable time $T_\varepsilon$ is defined as follows:

$$T_\varepsilon = \max\{\tau \mid \; \| \mathbf{M}_T(u_0) - u_T^t \|_2 \leq \varepsilon, 0 \leq T \leq \tau\}. \tag{1}$$

Since the true value $u_T^t$ cannot be obtained exactly, it is impossible to get $T_\varepsilon$ by solving nonlinear optimization problem (1). Inspired by the fact that the IC $u_0$ is often provided by an analysis field, and the associated analysis error can generally be controlled in a specified range, Mu et al. (2002) reduced the maximum predictable time problem to the following LBMPE problem.

If we have an estimation on the uncertainty of the IC as follows,

$$\left\| \boldsymbol{u}_0 - \boldsymbol{u}_0^t \right\| \leq \sigma , \tag{2}$$

then the LBMPE $T_l$ is defined as

$$T_l = \min_{\|\delta\boldsymbol{u}_0\| \leq \sigma} \{ T_{\boldsymbol{u}_0, \delta\boldsymbol{u}_0} \mid T_{\boldsymbol{u}_0, \delta\boldsymbol{u}_0} = \max \tau , \| \mathbf{M}_t(\boldsymbol{u}_0 + \delta\boldsymbol{u}_0) - \mathbf{M}_t(\boldsymbol{u}_0) \| \leq \varepsilon, 0 \leq t \leq \tau \} . \tag{3}$$

where $\sigma > 0$ denotes the accuracy of the IC in term of norm $\| \cdot \|$, $\delta\boldsymbol{u}_0$ is an initial perturbation superposed

on the IC. According to (2), the true initial state is within the constraint region, we have

$$T_l \leq T_\varepsilon .$$

Problem 2 The upper bound of maximum prediction error

When a forecast is produced from an incorrect initial IC $\boldsymbol{u}_0$, the prediction error at the prediction time $T$ is

$$E = \| \mathbf{M}_T(\boldsymbol{u}_0) - \boldsymbol{u}_T^t \| , \tag{4}$$

Similar to problem 1, since the true value $\boldsymbol{u}_T^t$ cannot be obtained precisely, Mu et al. (2002) instead introduced

the upper bound of the maximum prediction error within the given initial error limitation as follows:

$$E_u = \max_{\|\delta\boldsymbol{u}_0\| \leq \sigma} \| \mathbf{M}_T(\boldsymbol{u}_0 + \delta\boldsymbol{u}_0) - \mathbf{M}_T(\boldsymbol{u}_0) \| , \tag{5}$$

Note that $\boldsymbol{u}_0^t$ satisfies (2) and $\boldsymbol{u}_T^t = \mathbf{M}_T(\boldsymbol{u}_0^t)$ under the assumption of perfect model, we have

$$E \leq E_u .$$

Problem 3 The lower bound of maximum allowable initial error

Given the prediction time $T > 0$ and prediction precision $\varepsilon > 0$, the maximum allowable initial error is:

$$\sigma_{\max} = \max\{ \sigma \mid \| \mathbf{M}_T(\boldsymbol{u}_0 + \delta\boldsymbol{u}_0) - \boldsymbol{u}_T^t \| \leq \varepsilon, \| \delta\boldsymbol{u}_0 \| \leq \sigma \} . \tag{6}$$

Similar to problem 1 and 2, the above problem was reduced by Mu et al. (2002) to the following lower bound of

maximum allowable initial error:

$$\bar{\sigma}_{\max} = \max\{ \sigma \mid \| \mathbf{M}_T(\boldsymbol{u}_0 + \delta\boldsymbol{u}_0) - \mathbf{M}_T(\boldsymbol{u}_0) \| \leq \varepsilon, \| \delta\boldsymbol{u}_0 \| \leq \sigma \} . \tag{7}$$

## 2.2 Conditional nonlinear optimal perturbation (CNOP)

In consideration of the nonlinearity impacts, Mu et al. (2003) introduced CNOPs into the study of predictability

problems. Suppose the atmospheric or the oceanic motions can be described by the following dynamic system:

$$\begin{cases} \dfrac{\partial \boldsymbol{U}}{\partial t} + \boldsymbol{F}(\boldsymbol{U}, t) = 0, \\ \boldsymbol{U} \mid_{t=0} = \boldsymbol{U}_0, \end{cases} \tag{8}$$

where $U(x,t) = (U_1(x,t), U_2(x,t), \cdots, U_n(x,t))^T$ is the basic state, which is an $n$-dimension vector; the superscript $T$ represents transpose, $U_0$ is the initial basic state, $x = (x_1, x_2, \cdots, x_m)^T \in \Omega \subset \mathbf{R}^{\mathbf{m}}$ and $t$ are the spatial and temporal variables respectively; $t = 0$ is the initial time; and $F$ is a nonlinear partial differential operator.

Suppose $\mathbf{M}_\tau$ is the nonlinear transmission propagator from the initial time $t = 0$ to the forecast time $t = \tau$, thus the state of model (8) at $\tau$ time is

$$U(x, \tau) = \mathbf{M}_\tau(U_0), \tag{9}$$

If $u_0$ stands for the initial perturbation of the basic state $U(t)$ and $u_I(\tau)$ is the development of $u_0$ at time $\tau$, that is

$$u_I(\tau) = \mathbf{M}_\tau(U_0 + u_0) - \mathbf{M}_\tau(U_0), \tag{10}$$

then the initial perturbation $u_0^*$ is called the conditional nonlinear optimal perturbation (CNOP) if and only if $u_0^*$ is the solution of the following optimization problem:

$$J(u_0^*) = \max_{u_0 \in B_\sigma} \| \mathbf{M}_\tau(U_0 + u_0) - \mathbf{M}_\tau(U_0) \|. \tag{11}$$

where $B_\sigma = \{u_0 \mid u_0 \in \mathbf{R}^{\mathbf{n}}, \|u_0\| \leq \sigma\}$ is the constraint domain to the initial perturbation. In terms of the $L^2$-norm, $B_\sigma$ is a ball with the center at the origin and the radius $\sigma$. Besides, $J$ is called the objective function in the context of the optimal control theory.

**2.3 Estimation of the LBMPE**

Duan and Luo (2010) designed a numerical method to calculate the LBMPT in their research of predictability (Fig. 1). It should be noted that CNOPs stand for the initial uncertainty in the given constrain domain which leads to the largest prediction error. Therefore, the maximum prediction error can be estimated by solving CNOPs.

In detail, for a given first guess value $T_G$ of the prediction time, one can use a constraint nonlinear optimization algorithm to capture the CNOP so as to estimate the maximum prediction error $E_{T_G}$ at time $T_G$ caused by the initial error in a constraint domain $B_\sigma$.

If $E_{T_G} > E_m$ ( $E_m$ stands for the allowable prediction error), we try to reduce the integral time step $T_G = T_G - \Delta T$, where $\Delta T$ is a certain constant, and calculate the maximum prediction error at the reduced time.

If $E_{T_G} < E_m$, we will increase the integral time step $T_G = T_G + \Delta T$ and calculate the maximum prediction error at the new $T_G$.

If $T_G$ satisfies the conditions: $E_{T_G + \Delta T} > E_m$ and $E_{T_G - \Delta T} \leq E_m$, then $T_G$ is considered as the lower bound of maximum predictable time which satisfies the prediction precision $E_m$ under the given initial error. The operation flow chart is shown as Fig. 1.

**Figure 1.** Flow chart of solving the LBMPT

### 2.4 The two-dimensional Ikeda model

The following two-dimensional Ikeda model is adopted as the prediction model:

$$\begin{cases} x_1(t+1) = 1 + \mu(x_1(t)\cos\theta_t - x_2(t)\sin\theta_t) \\ x_2(t+1) = \mu(x_1(t)\sin\theta_t + x_2(t)\cos\theta_t) \end{cases}, \tag{12}$$

$$\theta_t = a - \frac{b}{1 + x_1(t)^2 + x_2(t)^2} \tag{13}$$

where $0 \leq \mu \leq 1$, $a = 0.4$ and $b = 6$.

From the expression of the model we find that the trigonometric functions appear in Equation (12), and Equation (13) is a fraction whose denominator includes two quadratic components. Thus, the two-dimensional Ikeda model has fairly strong nonlinearity (Li and Zheng, 2016). The solutions of model present different behaviors with the change of the model parameter $\mu$. When the parameter varies from 0 to 1, the numerical solutions change from a point attractor to periodic solutions, then to chaos, and end up with a limit cycle (Li and Zheng, 2016). The predictability problems are always launching under chaos. According to the conclusions given by Li and Zheng (2016), the model solution appeared chaos when $\mu \in [0.700, 0.902]$. Figure 2 shows the numerical solution of the last 5000 steps in 10000 integral steps while the initial value is set as $(x_0, y_0) = (0.25, -0.325)$ and the model parameter is $\mu = 0.83$.

**Figure 2.** The distribution of solutions at last 5000 steps when $\mu = 0.83$

### 3. The particle swarm optimization (PSO) algorithm

The particle swarm optimization (PSO) algorithm has recently got more and more attention (Eberhart et al., 2001; Banks et al., 2007, 2008; Poli et al., 2007). PSO algorithm was originally proposed by social psychologists

Kennedy and Eberhart (1995). It simulates the collective behavior about birds foraging. Each particle represents a potential solution in the PSO algorithm, flies with specific velocity respectively and adjust its trajectories according to the flying experiences of its own and the companion's, finally finds the optimal location in the solution space. To avoid the particles rapidly flying out the solving region and improve the ability of PSO searching global optimal solution, Shi and Eberhart (1998) introduced the maximum velocity and the inertia weight into PSOs, so as to restrain the particle's behaviors in the searching process. Clerc and Kennedy (2002) proposed a limiting factor for the two acceleration coefficients after they analyzed theoretically the convergence of the PSO.

The basic PSO algorithm consists of three processes, namely, generating particles' positions and velocities, assessing particles, and updating particles' positions and velocities.

Mathematical description of the classic PSO algorithm is as follows:

In an $n$ dimensional search space, each particle of the PSO represents a potential solution of the optimization problem. We denote $M$ the swarm size, $X_i(k) = \left( x_{i1}(k), x_{i2}(k), \cdots, x_{in}(k) \right)$ and $V_i(k) = \left( v_{i1}(k), v_{i2}(k), \cdots, v_{in}(k) \right)$ the position and the velocity of the $i$ th particle at the $k$th generation respectively, $P_i(k) = \left( p_{i1}(k), p_{i2}(k), \cdots, p_{in}(k) \right)$ the personal historical best position of the $i$ th particle found so far, $P_g(k) = \left( p_{g1}(k), p_{g2}(k), \cdots, p_{gn}(k) \right)$ the best position that the whole swarm attained so far, then particle $i$ 's velocity and position in the next $k+1$ generation can be updated according to the following formula:

$$v_{id}(k+1) = w v_{id}(k) + c_1 r_1 (p_{id}(k) - x_{id}(k)) + c_2 r_2 (p_{gd}(k) - x_{id}(k)) , \qquad (14)$$

$$x_{id}(k+1) = x_{id}(k) + v_{id}(k+1) , \qquad (15)$$

where $i = 1, 2, \cdots, M$ , $d = 1, 2, \cdots, n$ , $c_1$ and $c_2$ are the acceleration coefficients, which make particles having the ability to self-summarize and learn from excellent particles among the group to approach to own and group historical optimal point. $r_1$ and $r_2$ are two random numbers that subject to uniform distribution on interval [0,1]. $\omega$ is the inertia weight. It can be set as a fixed constant or a linear reduction function with the increase of the evolutional generations. The flow chart of the PSO algorithm is as follows:

**Figure 3.** The flow chart of the PSO algorithm.

When using a PSO to search CNOPs for the estimation of the LBMPT, the prediction error at the specified forecast time is the associated objective function $J$. The initial perturbation $\delta \boldsymbol{u}$, which is a two-dimensional vector in the search space in our situation, is the optimization variable.

## 4. Numerical experiments and their results analyses

### 4.1 The numerical experiments solving CNOPs by different optimization algorithms

In order to compare the performances of the ADJ-CNOP and PSO-CNOP in solving CNOPs, the CNOPs yielded by the filtering method are taken as the benchmark after fine dividing the constraint domain of initial perturbations. The filtering method is implemented as follows. The corresponding circumscribed square of a constraint region of the CNOP is considered, foursquare-meshes of a certain size are used to discretize the circumscribed square. For any mesh point outside the region, it is connected with the center of the region, the intersection point of this line with the boundary of the region is obtained. Integrating the Ikeda model from the initial basic state superimposed each of these intersection points and for the mesh points inside the region, the prediction error caused by each initial error can be obtained. CNOP refers to the mesh point which leads to the largest prediction error (Duan and Luo, 2010). Since the accuracy of the CNOP generated by the filtering method depends on the division size of the constraint region exclusively, the circumscribed square of the constraint ball of CNOPs are separated into $1001 \times 1001$ small quadrate patches with very small side length $1.6402 \times 10^{-5}$ in numerical experiments. For the detail description on the operation of the filtering method, one can refer to (Duan and Luo, 2010; Zheng et al., 2012).

In the numerical experiments, the initial basic state of the two-dimensional Ikeda model is $(x_0, y_0) = (0.25, -0.325)$, and model parameter is $\mu = 0.83$. The population size of the PSO $M = 60$, the maximum evolutional generation is set as 200, inertia weight $\omega = 0.729$ and accelerating factors $C_1 = 2.05$, $C_2 = 2.05$. The norms measuring IC errors and prediction errors are both $L^2$-norm, and the radius $\sigma$ of the constraint ball $B_\sigma$ is $8.201 \times 10^{-3}$.

The particle swarm initialization scheme in PSO-CNOP is as follows:

$\boldsymbol{X}_i = (x_{i,1}(0), x_{i,2}(0))$, are random vectors obeying uniform distribution on $B_\sigma$.

$\boldsymbol{V}_i(0) = (v_{i,1}(0), v_{i,2}(0)) = \boldsymbol{X}_i(0)$, $i = 1, 2, \cdots, M$.

The first guess of the perturbation $\delta u = (x_0, y_0)$ for ADJ-CNOP is randomly picked from $B_\sigma$. The constrained optimization algorithm used in the ADJ-CNOP is the SPG2.

With the prediction time increasing, there would appear simultaneously many CNOPs simultaneously for the two-dimensional Ikeda model because of the impact of the strong nonlinearity. Hence, different forecast times are adopted to test the ability of ADJ-CNOP and PSO-CNOP in attacking the nonlinearity obstacles. For each forecast time, numerical experiment using ADJ-CNOP or PSO-CNOP to obtain CNOPs is conducted 40 times respectively. 40 CNOPs are clustered by the fuzzy c-means clustering (FCM) method and the accuracy of the CNOPs is statistically analyzed.

The numerical experiment results show that when the integration time of prediction model is short, the corresponding objective function (11) presents a good behavior with the change of the initial perturbation. It has only two extreme values in the constraint ball of the initial perturbation. One is the global maximum and the extreme point corresponds to the global CNOP. The other is the local maximum and the extreme point is a local CNOP. Figure 4 shows the distribution of objective function values (OFV) when the prediction time are on 6th unit time steps (i.e., $6\Delta t$) (left) and $13\Delta t$ (right) respectively, in which the global maximum point locates at the point a, and the point b is the position of the local maximum.

**Figure 4.** The distributions of OFVs at the prediction time $6\Delta t$ (left) and $13\Delta t$ (right), in which dots a and b are the global and the local maximum points.

Tables 1 and 2 demonstrate the statistical analysis results of the CNOPs produced by ADJ-CNOP, PSO-CNOP when capturing CNOPs 40 times at the forecast time $6\Delta t$ and $13\Delta t$, and the related results generated by the filtering method. Through the FCM method, we find that CNOPs obtained by the ADJ-CNOP method are divided into two categories, one is related to the global CNOP that accounted for 47.5% (70%) for the forecast time $6\Delta t$ ($13\Delta t$) of the total, the other is the local CNOP that make up 52.5% (30%) of totality. However, 40 CNOPs captured by PSO-CNOP are completely the same and they are coincident with the CNOP yielded by the filtering method.

**Table 1.** Statistical analysis of CNOPs produced by different methods at $6\Delta t$

**Table 2.** Same as Table 1 except at $13\Delta t$

From tables 1 and 2, we can see that although the prediction time is short, the ADJ-CNOP method still has large probability to capture local CNOPs, while PSO-CNOP can always catch the global CNOP. Actually, we can draw the same conclusion with the prediction time being increased to $13\,\Delta t$.

When the forecast time increases to $14\,\Delta t$, the 40 CNOPs yielded by ADJ-CNOP and PSO-CNOP are demonstrated in the following Fig. 5.

**Figure 5.** The distributions of OFVs at the prediction time $14\,\Delta t$, where dots in the left (right) panel are the CNOPs produced by ADJ-CNOP (PSO-CNOP).

Figure 6 indicates all CNOPs generated by the filtering method with fine division of the constraint domain of initial perturbations (the circumscribed square of the constraint ball of CNOPs are separated into $1001\times1001$ small quadrate patches with very small side length $1.6402\times10^{-5}$).

**Figure 6.** The distributions of OFVs at the prediction time $14\,\Delta t$, where dots are the CNOPs produced by the filtering method.

Figure 6 shows that when the prediction time reaches to $14\,\Delta t$, there exists many global CNOPs and all of them are located in a line. Based on which, we find that ADJ-CNOP not only get global or local CNOPs, but also capture false CNOPs at the prediction time $14\,\Delta t$. Since no matter how small an area we take around one of these "CNOPs", there always exists one point whose objective function value is larger than the objective function value of the "CNOP". According to the definition of CNOPs, these "CNOPs" are not true CNOPs. Hence, we call them false CNOPs.

Additionally, comparing the right panel of Fig. 5 with Fig. 6, it is easy to know that although many CNOPs are produced by PSO-CNOP in 40 repeated numerical experiments, all of the CNOP points are located in the same line as the one presented by the filtering method. Therefore, the CNOPs yielded by PSO-CNOP are all global.

When we keep extending the prediction time, behavior of the objective function will get much worse, and there will appear more extreme points. In order to verify the performance of the PSO-CONP method for solving CNOPs in strong nonlinear case, the mean value and variance of the OFVs of the 40 CNOPs at different forecast time are calculated and compared with the maximal OFV (MOFV) obtained by the filtering method.

**Table 3.** The precision analysis of the CNOPs produced by PSO-CNOP at different forecast times

According to table 3, OFVs of 40 CNOPs calculated by the PSO-CNOP method at each forecast time are almost consistent with the maximum of the objective function gotten by the filtering method at the same forecast time. Therefore, PSO-CNOP is still capable of solving global CNOPs of the two-dimensional Ikeda model for long forecast time. Figure 7 demonstrate the distributions of OFVs at the prediction time $15\,\Delta t$, $18\,\Delta t$ and $22\,\Delta t$, as well as the locations of all CNOPs generated by PSO-CNOP.

**Figure 7.** The distribution of OFVs at the prediction time $15\,\Delta t$ (the left of upper panel), $18\,\Delta t$ (the right of upper panel) and $22\,\Delta t$ (the lower panel), where dots denote CNOPs captured by PSO-CNOP.

From the upper two panels of Fig. 7, we can see clearly that the CNOPs are all located in the maximal OFV region, which indicates that all of the CNOPs captured by the PSO- CNOP method are global. Because of the complexity of the distributions of OFVs at the prediction time $22\,\Delta t$, it cannot be confirmed directly from the lower panel of Fig. 7 whether or not the CNOPs are global. Therefore, we select one CNOP randomly from each cluster of the 40 CNOPs and zoom in the graph nearby the CNOP point to look into the OFV distribution. Figure 8 gives one of the results, from which we can see that the CNOP still locates in maximal OFV area.

**Figure 8.** Local distribution of the OFV at the prediction time $22\,\Delta t$ nearby the CNOP point.

With further increasing of the prediction time, the strong nonlinearity deteriorates the behavior of the objective function seriously. In this situation, predictability study based on the CNOP method becomes no longer meaningful because the CNOPs are too dispersive.

**4.2 Comparison between PSO-CNOP and GA-CNOP**

In the following, the GA is adopted to capture CNOPs of the two-dimensional Ikeda model, and the results are compared with the ones obtained by the PSO-CNOP. The method using the GA to compute CNOP is called GA-CNOP. The relevant operation flow chart of the GA is showed in Fig. 9.

**Figure 9.** The flow chart of the GA.

The configuration of the genetic operators and the relevant parameter are same as in (Zheng et al. 2014). More detailed description of the GA and numerical experiment scheme of GA-CNOP can refer to (Zheng et al. 2014). The performance of PSO-CNOP and GA-CNOP in solving the CNOPs are tested for different population size, the results are statistically analyzed and presented in Tables 4 and 5.

**Table 4.** Mean CFV of 40 CNOPs produced by PSO-CNOP for every given population size

**Table 5.** Same as Table 4 except 40 CNOPs are produced by GA-CNOP

It can clearly be seen from Tables 4 and 5 that the optimal population size of PSO-CNOP is about 15, but the optimal population size of GA-CNOP is about 40. Furthermore, the computational times and CFV calculated respectively by PSO-CNOP with a population size of 15, and GA-CNOP with a population size of 40, are compared with that of ADJ-CNOP. Table 6 illustrate the mean CFV and the average computation time obtained by different methods in their 40 numerical experiments.

**Table 6.** Analysis of CFV and average computational time by different methods

From the numerical results we can see that CFV of GA-CNOP is almost the same with PSO-CNOP, the GA is an effective optimal algorithm to obtain the optimal solution. However, the GA is more time consuming than the PSO. At the same time, the computational time of PSO-CNOP and GA-CNOP is much greater than ADJ-CNOP, which is also the difference between stochastic searching algorithms and deterministic searching algorithms. Fortunately, in intelligent optimization algorithms, the parallel computation can be easily realized. The operators of different individuals in one generation are independent, and can be done in different CPUs, which thereby can take full advantage of fast developed parallel computation technology (Fang et al., 2009).

**4.3 Estimation of the LBMPT**

The CNOP method can be adopted to do similar study on other two predictability sub-problems. Here we only focus on the estimation of the LBMPT to discuss the influence of nonlinearity. To demonstrate the effectiveness of PSO-CNOP in solving this problem, the filtering method, ADJ-CNOP and PSO-CNOP are used in the numerical experiments respectively. In order to compare the estimation accuracy of the LBMPT generated by ADJ-CNOP and PSO-CNOP, the LBMPT computed by the filtering method with fine division is taken as the benchmark. Different allowable prediction errors $E_m$: 0.5, 0.8, 1.1, 1.4, 1.7 and different constrained radius $\delta$ of initial perturbations: 0.01, 0.02, 0.03, 0.04, 0.05 are employed to verify the performance of the ADJ-CNOP and PSO-CNOP method. The lattice spacing of the filtering method is 0.001. Tables 7, 8 and 9 illustrate the LBMPTs computed by the filtering method, PSO-CNOP and ADJ-CNOP respectively.

**Table 7.** The LBMPT computed by the filtering method.

**Table 8.** The LBMPT estimated by the PSO-CNOP method.

Compare table 7 with table 8, we find that the LBMPTs estimated by the PSO-CNOP approach is completely same as the ones computed by the filtering method. It's worthy to mention that with the prediction time extending, although the objective function have multiple extreme values and PSO-CNOP would produce

different CNOPs in different numerical experiments, each one of these CNOPs can be used to estimate the maximum prediction error, and the final LBMPTs obtained are same since they are all global.

**Table 9.** The LBMPT estimated by the ADJ-CNOP method.

The bold numbers in table 9 are the LBMPTs that are different from the ones computed by the filtering method. The LBMPTs yielded by the ADJ-CNOP method are generally larger. The reason is that the CNOP given by the

ADJ-CNOP method is often local, even false. Therefore the estimation of the maximum prediction error based on the CNOP is usually questionable and untrusted.

To investigate the probability that the ADJ-CNOP method generates incorrect LBMPTs, we operate the numerical experiment shown in table 9 40 times with various first guesses of the initial perturbations. The statistical analysis results are given in table 10.

**Table 10.** Incorrect ratio in total 40 LBMPTs yielded by ADJ-CNOP.

From table 10, we can see that the ratio of incorrect LBMPTs based on ADJ-CNOP is high. The highest one even reaches 95% when $\delta = 0.02$ and $E_m = 1.7$. This problem is serious for real weather forecasts since it can mislead forecasts with large probability.

## 5. Conclusion and Discussion

Since the two-dimensional Ikeda model has strong nonlinearity, when we utilize the ADJ-CNOP method to capture CNOPs, not only global or local CNOPs, but even false CNOPs are obtained. The reason for which is that in the case of strong nonlinearity, the gradient provided by the adjoint model is incorrect. When the traditional optimization algorithm uses a wrong descent direction to search extreme values of the objective function, false CNOPs are presented.

The PSO is a heuristic search algorithm based on population. It can overcome the nonlinear influences and produce global CNOPs with high probability. In addition, the operation of the PSO algorithm is simple. This study applies the PSO algorithm to capture the CNOP of the two-dimensional Ikeda model. Numerical experiment results with different forecast times demonstrate that although the objective function has awful behavior and multiple extreme values, PSO-CNOP still can capture global CNOPs.

Furthermore, precision problems of using ADJ-CNOP to estimate the LBMPT are investigated. Results show that when the objective function has multiple extreme values, ADJ-CNOP has large probability to produce the local CNOP, hence induce false estimation of the LBMPT. As PSO-CNOP can always yield global CNOPs, therefore, the estimation of the LBMPT presented by PSO-CNOP is precise. It is consistent with the one yielded by the filtering method with fine division.

As we know, our numerical experiments are focusing on two-dimensional prediction model only. When considering high dimensional and more complex model, if the classic PSO algorithm used in this study can overcome the influence of high dimension and the computation time can meet the real requirement is still unknown. The problems of curse of dimensionality and multimodal are big challenge for almost all intelligent optimization algorithms, also the PSO. Whether it can be effective in higher dimensional and more complicated model deserves further research. In short, the PSO-CNOP approach is an alternative method to study predictability problems in the case of the strong nonlinearity.

*Acknowledgements.* This work was supported by the National Natural Science Foundation of China (Grant Nos. 41430426 and 41331174).

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

**Table 1.** Statistical analysis of CNOPs produced by different methods at 6 $\Delta t$

| Method | OFV | CNOP | Proportion |
|---|---|---|---|
| Filtering | 3.7533E-2 | (-3.5066E-3, 7.4131E-3) | |
| ADJ-CNOP | 3.7533E-2 | (-3.5068E-3, 7.4130E-3) | 47.5% |
| | 2.3973E-2 | ( 2.9072E-3, -7.6683E-3) | 52.5% |
| PSO-CNOP | 3.7533E-2 | (-3.5068E-3, 7.4130E-3) | 100% |

**Table 2.** Same as Table 1 except at 13 $\Delta t$

| Method | OFV | CNOP | Proportion |
|---|---|---|---|
| Filtering | 1.5301 | (-3.4679E-3, 7.4313E-3) | |
| ADJ-CNOP | 1.5301 | (-3.4682E-3, 7.4311E-3) | 70% |
| | 0.8516 | (-1.9285E-3,-7.9706E-3) | 30% |
| PSO-CNOP | 1.5301 | (-3.4682E-3, 7.4311E-3) | 100% |

**Table 3.** The precision analysis of the CNOPs produced by PSO-CNOP at different forecast times

| Prediction time | MOFV of the filtering method | CFV of PSO-CNOP | |
|---|---|---|---|
| | | Mean value | Variance |
| 15 $\Delta t$ | 1.6106 | 1.6106 | 3.3142E-16 |
| 16 $\Delta t$ | 1.3052 | 1.3052 | 8.6189E-14 |
| 17 $\Delta t$ | 1.6521 | 1.6521 | 2.0092E-16 |
| 18 $\Delta t$ | 1.7807 | 1.7807 | 7.6313E-17 |
| 19 $\Delta t$ | 1.5401 | 1.5401 | 1.7300E-16 |
| 20 $\Delta t$ | 1.3482 | 1.3482 | 8.8314E-10 |
| 21 $\Delta t$ | 1.6980 | 1.6980 | 3.3546E-10 |
| 22 $\Delta t$ | 1.6602 | 1.6593 | 4.6798E-07 |

**Table 4.** Mean CFV of 40 CNOPs produced by PSO-CNOP for every given population size.

| Prediction time | Population size | | | | | |
|---|---|---|---|---|---|---|
| | 5 | 10 | 15 | 30 | 45 | 60 |
| 13 $\Delta t$ | 1.5301 | 1.5301 | 1.5301 | 1.5301 | 1.5301 | 1.5301 |
| 19 $\Delta t$ | 1.5209 | 1.5396 | 1.5401 | 1.5401 | 1.5401 | 1.5401 |

**Table 5.** Same as Table 4 except 40 CNOPs are produced by GA-CNOP

| Prediction time | Population size | | | | | |
|---|---|---|---|---|---|---|
| | 16 | 26 | 36 | 40 | 50 | 60 |
| 13 $\Delta t$ | 1.5299 | 1.5300 | 1.5300 | 1.5300 | 1.5300 | 1.5300 |
| 19 $\Delta t$ | 1.5019 | 1.5295 | 1.5357 | 1.5400 | 1.5400 | 1.5400 |

**Table 6.** Analysis of CFV and average computational time by different methods

| Prediction time | 6 $\Delta t$ | | 13 $\Delta t$ | |
|---|---|---|---|---|
| Method | Mean CFV | Time (s) | Mean CFV | Time (s) |
| ADJ-CNOP | 3.0100E-2 | 5.0000E-4 | 1.3944 | 6.7500E-4 |
| PSO-CNOP | 3.7533E-2 | 0.0032 | 1.5301 | 0.0058 |
| GA-CNOP | 3.7533E-2 | 0.0299 | 1.5300 | 0.0358 |

**Table 7.** The LBMPT computed by the filtering method

| $E_m$ | $\delta$ =0.01 | $\delta$ =0.02 | $\delta$ =0.03 | $\delta$ =0.04 | $\delta$ =0.05 |
|---|---|---|---|---|---|
| 0.5 | 9 | 7 | 7 | 6 | 6 |
| 0.8 | 10 | 9 | 7 | 7 | 6 |
| 1.1 | 10 | 10 | 9 | 7 | 7 |
| 1.4 | 10 | 10 | 9 | 7 | 7 |
| 1.7 | 12 | 12 | 12 | 11 | 10 |

**Table 8.** The LBMPT estimated by the PSO-CNOP method

| $E_m$ | $\delta$ =0.01 | $\delta$ =0.02 | $\delta$ =0.03 | $\delta$ =0.04 | $\delta$ =0.05 |
|---|---|---|---|---|---|
| 0.5 | 9 | 7 | 7 | 6 | 6 |
| 0.8 | 10 | 9 | 7 | 7 | 6 |
| 1.1 | 10 | 10 | 9 | 7 | 7 |
| 1.4 | 10 | 10 | 9 | 7 | 7 |
| 1.7 | 12 | 12 | 12 | 11 | 10 |

**Table 9.** The LBMPT estimated by the ADJ-CNOP method

| $E_m$ | $\delta$ =0.01 | $\delta$ =0.02 | $\delta$ =0.03 | $\delta$ =0.04 | $\delta$ =0.05 |
|---|---|---|---|---|---|
| 0.5 | 9 | 7 | 7 | **7** | **7** |
| 0.8 | 10 | 9 | **8** | 7 | 6 |
| 1.1 | 10 | **11** | 9 | **9** | **8** |
| 1.4 | 10 | 10 | **10** | **9** | **9** |
| 1.7 | 12 | **13** | 12 | 11 | 10 |

**Table 10.** Incorrect ratio in total 40 LBMPTs yielded by ADJ-CNOP

| $E_m$ | $\delta$ =0.01 | $\delta$ =0.02 | $\delta$ =0.03 | $\delta$ =0.04 | $\delta$ =0.05 |
|---|---|---|---|---|---|
| 0.5 | 15% | 40% | 32.5% | 52.5% | 45% |
| 0.8 | 22.5% | 55% | 35% | 42.5% | 50% |
| 1.1 | 7.5% | 42.5% | 35% | 42.5% | 57.5% |
| 1.4 | 25% | 32.5% | 35% | 52.5% | 50% |
| 1.7 | 12.5% | 95% | 30% | 60% | 60% |

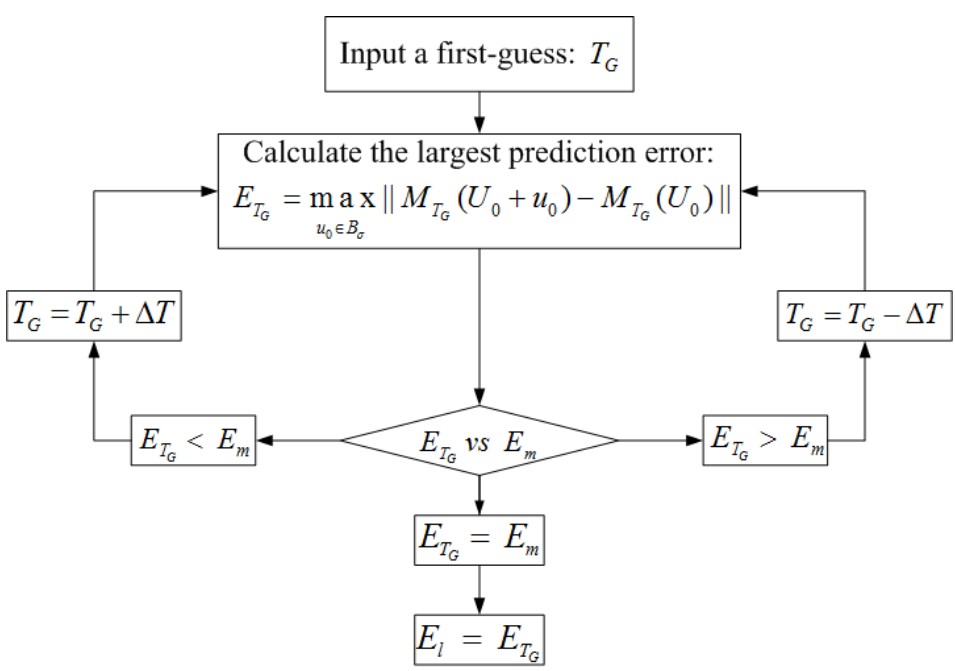

**Figure 1.** Flow chart of solving the LBMPT.

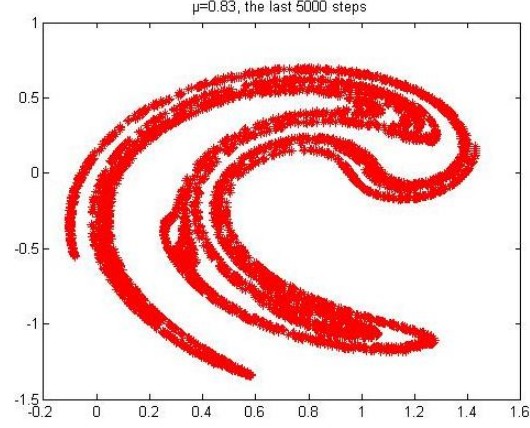

 **Figure 2.** The distribution of solutions at last 5000 steps when $\mu = 0.83$

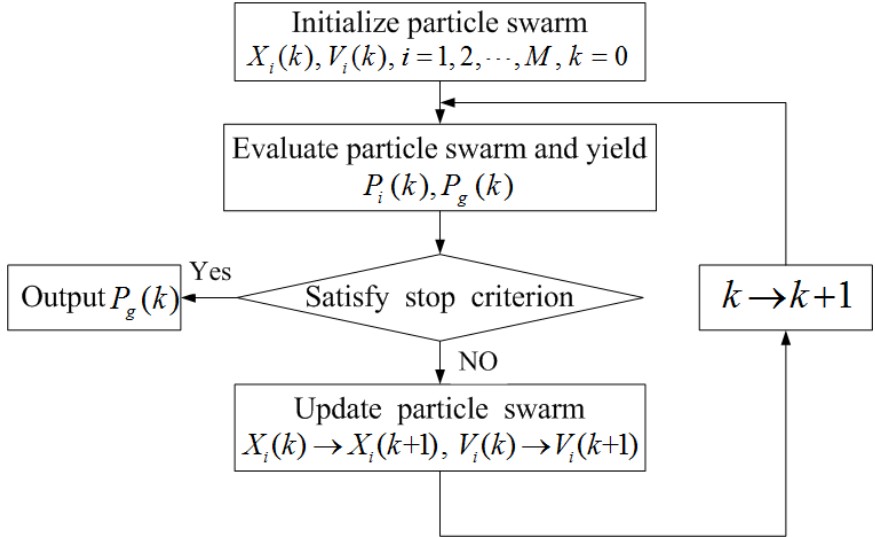

**Figure 3.** The flow chart of the PSO algorithm.

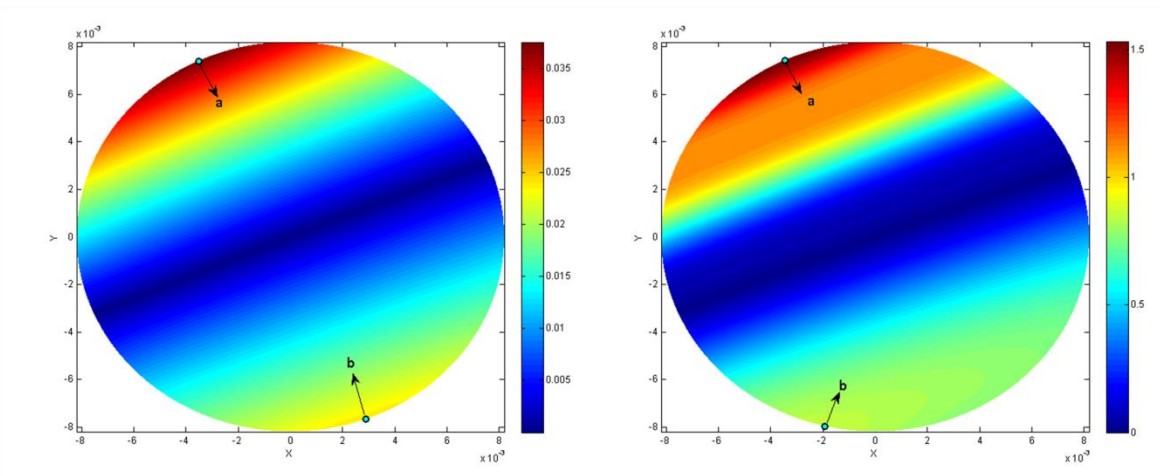

5          **Figure 4.** The distributions of OFVs at the prediction time $6\,\Delta t$ (left) and $13\,\Delta t$ (right),

in which dots a and b are the global and the local maximum points.

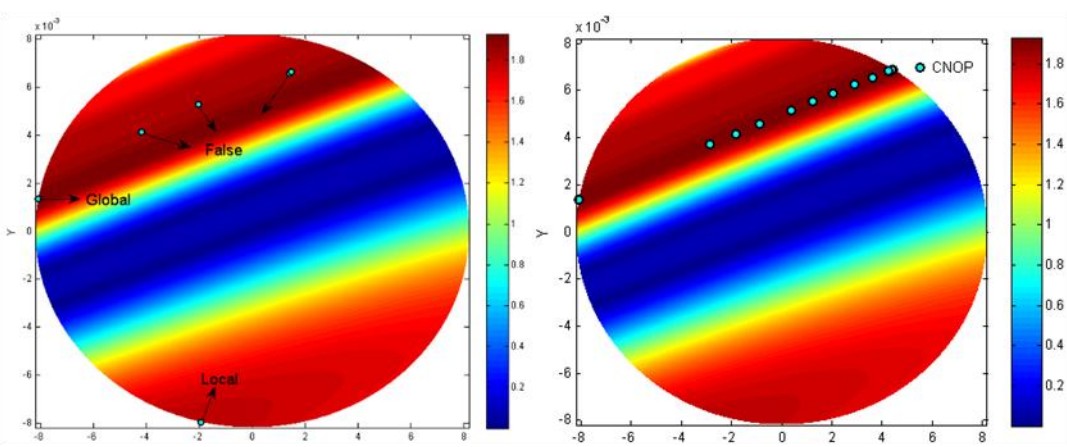

**Figure 5.** The distributions of OFVs at the prediction time $14\,\Delta t$, where dots in the left (right) panel are the CNOPs

produced by ADJ-CNOP (PSO-CNOP).

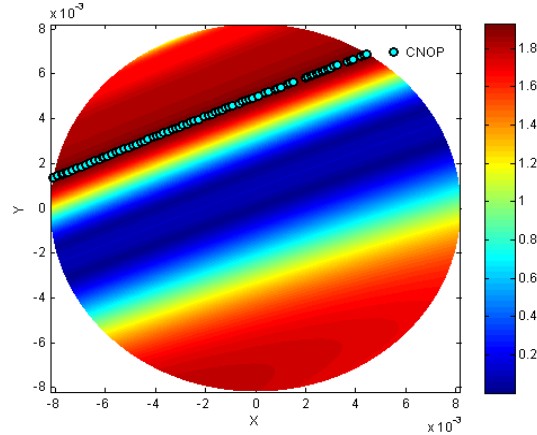

**Figure 6.** The distributions of OFVs at the prediction time $14\,\Delta t$, where dots are the CNOPs produced by the filtering

method.

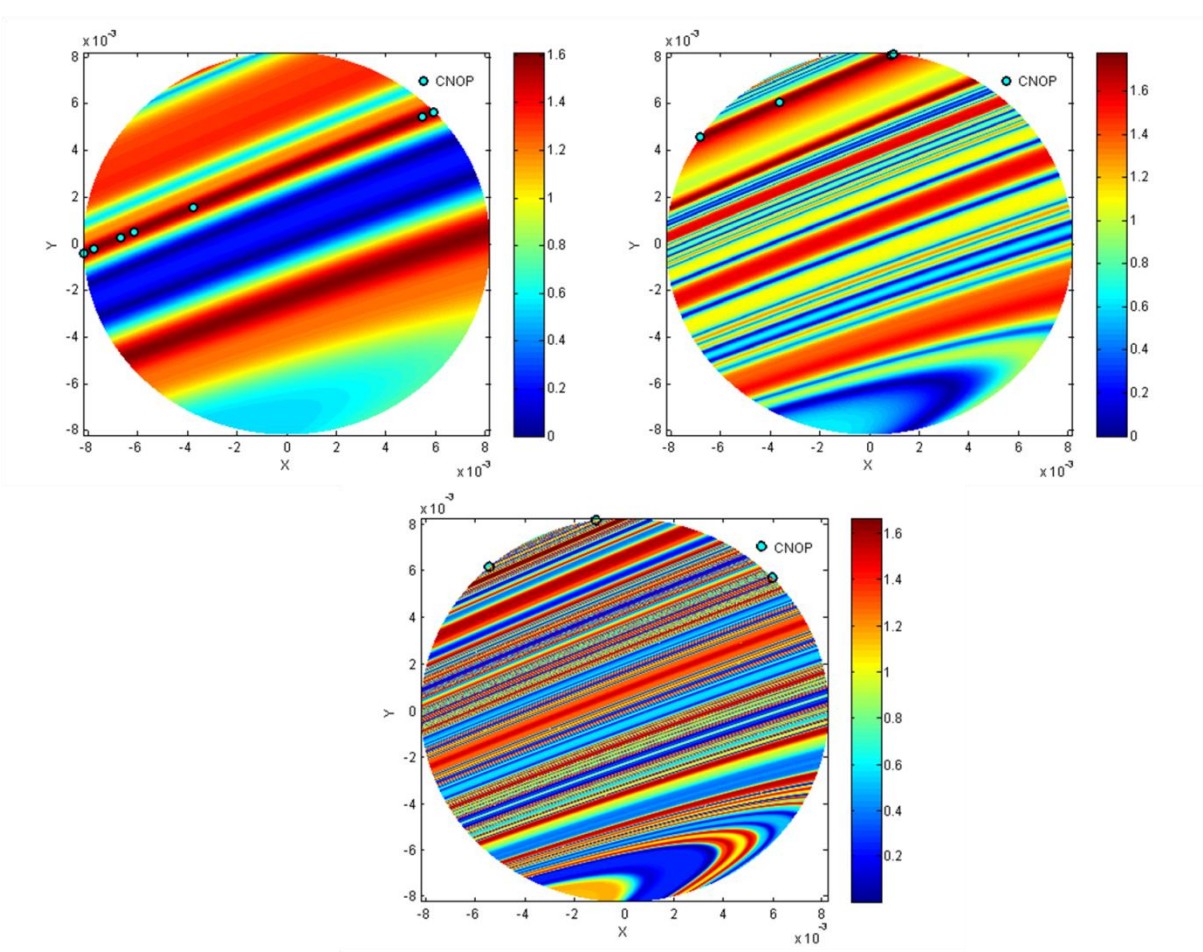

**Figure 7.** The distribution of OFVs at the prediction time 15 $\Delta t$ (the left of upper panel), 18 $\Delta t$ (the right of upper panel) and 22 $\Delta t$ (the lower panel), where dots denote CNOPs captured by PSO-CNOP.

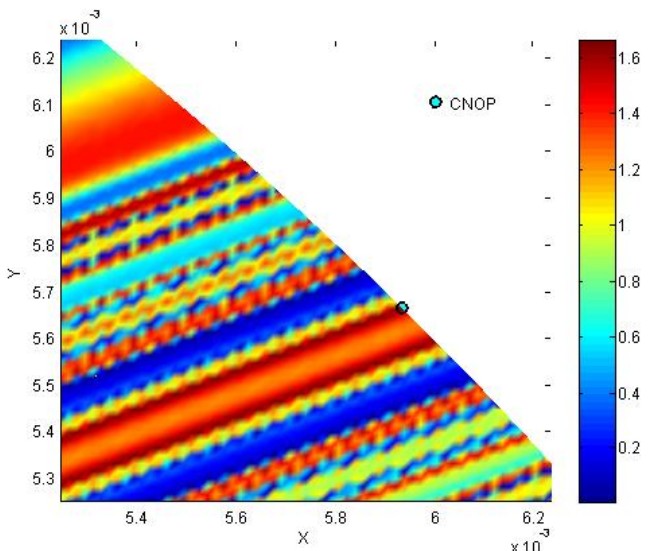

**Figure 8.** Local distribution of the OFV at the prediction time 22 $\Delta t$ nearby the CNOP point.

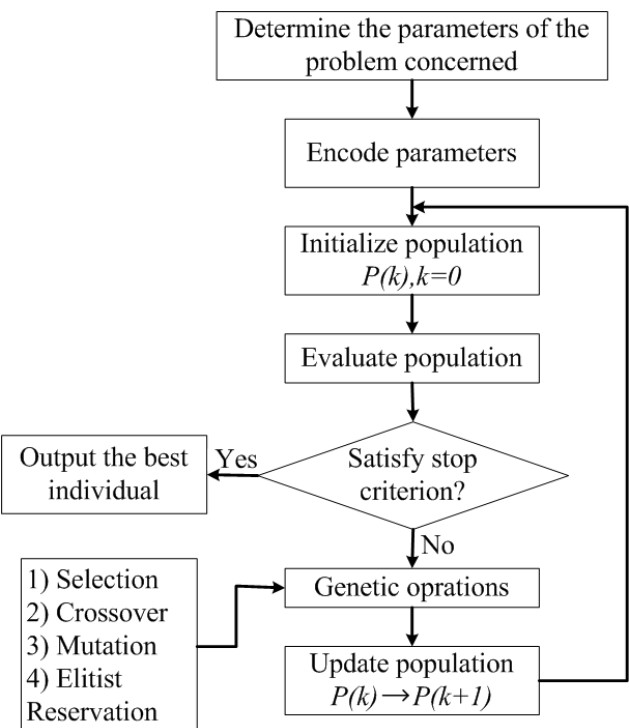

**Figure 9.** The flow chart of the GA.