# Peer review of "Conditional nonlinear optimal perturbations based on the particle swarm optimization and their applications to the predictability problems"

_Nonlinear Processes in Geophysics, 2016_

## Referee Comment (RC1) · Anonymous Referee #1 · 21 Dec 2016

General comment

The authors applied the particle swarm optimization (PSO) algorithm to solve the conditional nonlinear optimal perturbation (CNOP) and the lower bound of maximum predictable time (LBMPT). The results obtained by the PSO algorithm were compared to those by the traditional optimization algorithm (such as, a gradient descent algorithm based on the adjoint model, ADJ). The authors found that the PSO algorithm had advantage to compute the CNOP when the initial perturbation was large or the prediction time was long for the strong nonlinearity of the dynamical model on the prediction variable. Authors attempted to obtain the CNOP using the PSO algorithm. Considering the applications of CNOP, it is an interesting work.

Major remarks (1): Please state the advantages of the PSO algorithm in detail, especially comparing to the ADJ algorithm. On the other hand, the genetic algorithm (GA) was also applied to compute the CNOP by authors. And, the authors have published some related articles (Zheng et al. 2012, 2014). The authors should explain the difference between PSO and GA in detail for readers. It is better to compare the numerical results with the ones by using the three methods (PSO, GA, and ADJ algorithms) in the revised manuscript.

(2): The initial guess population size of the PSO algorithm is 60 for 40 optimization processes. The size is enough for the PSO algorithm. However, how many the number of the initial guess values is for the ADJ algorithm? 40 optimization processes and an initial guess values? Is it fair to obtain the CNOP using the PSO and ADJ algorithm? In general, the global CNOP could be obtained by choosing a number of initial guess values using the ADJ algorithm. It is important to choose the initial guess values using the optimization method. The computation results of the CNOP could be divided into three types using a number of initial guess values for the ADJ algorithm. The first one is the global CNOP. The second one is the local CNOP. The last one fails to compute the optimal value. In table 1 and 2, the authors showed the proportion of the global CNOP and local CNOP using 40 times. It is reasonable and not shortcoming for calculating the CNOP using the ADJ algorithm. It is acceptable to obtain the global CNOP without all initial guess values.

Specific remarks:

(1): Page 8, line 27-Page 9, line 1: Please introduce how to separate small quadrate patches. (2): Page 9, line 5-8: "The population size of the PSO $M = 60$, the maximum evolutional generation is set as 200, inertia weight $=0.729$ ïĄů and accelerating factors $c1=2.05,c2=2.05$. The norms measuring IC errors and prediction errors are both 2 L - norm, and the radius of the constraint ball is $8.201*10\hat{\,}(-3)$". Please explain the reasons to choosing of the parameter. Whether are the results dependent on the choosing of the parameter or not? (3): Page 9, line 14: The authors emphasized the importance of the

impact of the strong nonlinearity. Please tell the readers that how to define the strong nonlinearity for the dynamical system. (4): The computational cost is considered as the index to calculate the optimal value. The computational cost (such as computation time, iteration times) should be shown by the PSO, GA and ADJ algorithms. (5): When the optimization time increases, please show the numerical results of computing CNOP being similar to Table 1 and 2 (6): The value of the PSO algorithm is that it is applicable to obtain the CNOP in high dimensional and more complex model. In the section of discussion, this issue should be further discussed. (7): Which ADJ algorithm is applied in the manuscript? The spectral projected gradient (SPG), the sequential quadratic programming (SQP), limited memory BFGS (L-BFGS)? The algorithm employed in the manuscript should be introduced. Please show the references of these algorithms.

---

## Referee Comment (RC2) · Anonymous Referee #2 · 21 Dec 2016

For a given dynamical system the conditional nonlinear optimal perturbations (CNOP), which cause the largest prediction error at the prediction time, is one of the keys to estimate the lower bound of maximum predictable time in the predictability problem. This paper introduces the particle swarm optimization (PSO), an intelligence algorithm, to compute the CNOP. A simple two dimensional Ikeda model is employed and its CNOPs are calculated using both the traditional adjoint approach, and the PSO algorithm. Numerical results show that both approaches can the global CNOPs in the presence of small initial perturbation with short prediction period, and the adjoint approach may also get some local CNOPs. However, when the initial perturbation become large or the prediction period extended, the strong nonlinearity of the dynamical model may

lead to the failure of the adjoint method to get the global CNOP. The PSO approach still work effectively to get the global CNOPs with high probability if the population is large. These results indicate that the PSO maybe also be an advanced and effective tool to compute the CNOP for a more complex dynamical system like the weather forecast system in the future. The paper is well organized and all the tables and figures are of good quality and well used. Therefore, it is recommended to publish the paper with minor modification.

Following are some suggestion for modification of the paper 1. In the numerical experiments, the filtering method are taken as the benchmark. It is better to describe what the filtering method is and its concrete steps.Âă 2. There are several parameters to tune when the PSO approach is used. The authors should better indicate if the results are sensitive to these parameters, like the initial velocity, and the inertial weight. 3. The authors also mentioned the generic algorithm GA). If possible, a simple comparison of this approach to the PSO would help the reader a lot to know the advantage of the PSO approach.

Following are some minor corrections of the paper 1. P1 Ln10: the CNOP method has been . . . should be the CNOP has been. . . 2. P2 Ln13: Kalnay, 2003 should be Kalnay 2003. 3. P5 Ln 3: T also depends on u0, , . 4. P7 Ln7-8: The Ikeda model is better to rewrite in (x1, x2) instead of (x,y) to make it compatible with note on P8 Ln7. 5. P10L3: are completely same . . . should be are completely the same . . .

---

## Author Comment (AC1) · 18 Jan 2017

Authors' Responses We are very grateful to the anonymous reviewers for their constructive comments and suggestions that have helped us improve the manuscript (npg-2016-55). We have modified our manuscript according to your comments and suggestions in the revised manuscript. In the following, we reply all the comments and suggestions: Comments of Reviewer #1: C1. Please state the advantages of the PSO algorithm in detail, especially comparing to the ADJ algorithm. On the other hand, the genetic algorithm (GA) was also applied to compute the CNOP by authors. And, the authors have published some related articles (Zheng et al. 2012, 2014). The authors should explain the difference between PSO and GA in detail for readers. It

is better to compare the numerical results with the ones by using the three methods (PSO, GA, and ADJ algorithms) in the revised manuscript. A1. Thank you very much for your useful suggestions! We have added the comparison between the PSO and the ADJ method, and discussed the difference between the PSO and the GA in the revised manuscript. Please See Page 3, lines 14-21. The comparison numerical experiments with PSO, GA, and ADJ algorithms have been operated and the results are showed in the revised manuscript. Please see Page 12, lines 11-26. Page 13, lines 1-7. Page 20. C2. The initial guess population size of the PSO algorithm is 60 for 40 optimization processes. The size is enough for the PSO algorithm. However, how many the number of the initial guess values is for the ADJ algorithm? 40 optimization processes and an initial guess values? Is it fair to obtain the CNOP using the PSO and ADJ algorithm? A2. The PSO is a population-based heuristic optimal algorithm, it start with a set of potential solutions (i.e., the initial population), here 60 is the specified population size of the PSO in our numerical experiments. ADJ-CNOP is a deterministic optimal algorithm, there is one initial guess value in the search process. For statistical analysis, we conducted 40 times numerical experiments using ADJ-CNOP and PSO-CNOP respectively. For ADJ-CNOP, 40 times numerical experiments are performed with 40 different initial guess value. For PSO-CNOP, 40 times numerical experiments are conducted with 40 initial population. Therefore, the comparison scheme between PSO-CNOP and ADJ-CNOP is fair. C3. Page 8, line 27-Page 9, line 1: Please introduce how to separate small quadrate patches. A3. Thank you very much for your kind suggestions! We have explained in detail how to separate small quadrate patches in the revised manuscript. Please see Page 9, lines 4-10. C4. Page 9, line 5-8: "The population size of the PSO M = 60, the maximum evolutional generation is set as 200, inertia weight =0.729 and accelerating factors c1=2.05, c2=2.05. The norms measuring IC errors and prediction errors are both 2 L-norm, and the radius of the constraint ball is 8.201*10Ë̈Ę(-3)". Please explain the reasons to choosing of the parameter. Whether are the results dependent on the choosing of the parameter or not? A4. Inertia weight =0.729 and accelerating factors

$c_1=2.05$, $c_2=2.05$ are commonly used parameter values for the PSO (Clerc, M., and J. Kennedy., 2002; Banks et al., 2007, 2008). Considering both the computational time and the optimal precision, we take the population size of the PSO as 60, and the maximum evolutional generation as 200. The radius of the constraint ball is set by the method given in (Mu and Zhang, 2006). The results are dependent on the choosing of inertia weight and accelerating factors. Fortunately, 0.729, 2.05 and 2.05 are effective for general optimization problems. In addition, it seems that there is no general set rules of population size and maximum evolutional generation that are suitable for any optimization problems. C5. Page 9, line 14: The authors emphasized the importance of the impact of the strong nonlinearity. Please tell the readers that how to define the strong nonlinearity for the dynamical system. A5. Thank you very much for your helpful suggestions! We have added the explanation of the strong nonlinearity of the Ikeda model. Please see Page 7, lines 10-12. C6. The computational cost is considered as the index to calculate the optimal value. The computational cost (such as computation time, iteration times) should be shown by the PSO, GA and ADJ algorithms. A6. Thank you very much for your useful suggestions! We have conducted the comparison numerical experiment of computational cost, and the results have been added in the revised manuscript. Please see Page 12, lines 18-26. Page 13, lines 1-7. Page 20. C7. When the optimization time increases, please show the numerical results of computing CNOP being similar to Table 1 and 2. A7. When the prediction time increases, the objective function has multiple extreme values, the types of CNOP also increase, after the comparison with the CNOP calculated by the filtering method, we obtain the statistical analysis of CNOPs produced by PSO-CNOP and ADJ-CNOP at different time respectively. CNOPs Prediction time Method Global Local False 16 PSO-CNOP 100% 0 0 ADJ-CNOP 27.5% 5% 67.5% 18 PSO-CNOP 100% 0 0 ADJ-CNOP 32.5% 17.5% 50% 20 PSO-CNOP 100% 0 0 ADJ-CNOP 5% 65% 30% Since no matter how small an area we take around one of these "CNOPs", there always exists one point whose objective function value is larger than the objective function value of the "CNOP". According to the definition of CNOPs, these "CNOPs"

are not true CNOPs. Hence, we call them false CNOPs. C8. The value of the PSO algorithm is that it is applicable to obtain the CNOP in high dimensional and more complex model. In the section of discussion, this issue should be further discussed. A8. Thank you very much for your helpful suggestions! The issues whether the PSO is applicable to obtain the CNOP in high dimensional and more complex model are further discussed in the revised manuscript. Please see Page 15, lines 4-6. C9. Which ADJ algorithm is applied in the manuscript? The spectral projected gradient (SPG), the sequential quadratic programming (SQP), limited memory BFGS (L-BFGS)? The algorithm employed in the manuscript should be introduced. Please show the references of these algorithms. A9. Thank you very much for your useful suggestions! The algorithm employed in ADJ-CNOP have been added it in the revised manuscript. Please see Page 9, lines 23-24. The references of these algorithms have been added in the revised manuscript. Please see Page 15, lines 16-17. Page 16, lines 18-19. Page 17, lines 19-20.

Please also note the supplement to this comment:
http://www.nonlin-processes-geophys-discuss.net/npg-2016-55/npg-2016-55-AC1-supplement.zip

---

## Author Comment (AC2) · 18 Jan 2017

Authors' Responses We are very grateful to the anonymous reviewers for their constructive comments and suggestions that have helped us improve the manuscript (npg-2016-55). We have modified our manuscript according to your comments and suggestions in the revised manuscript. In the following, we reply all the comments and suggestions: Comments of Reviewer #2: C1. In the numerical experiments, the filtering method are taken as the benchmark. It is better to describe what the filtering method is and its concrete steps. A1. Thank you very much for your useful suggestions! We have added this content in the revised manuscript. Please see Page 9, lines 4-10. C2. There are several parameters to tune when the PSO approach is used.

[Figure]

The authors should better indicate if the results are sensitive to these parameters, like the initial velocity, and the inertial weight. A2. The initial velocity often sets at about 10-20% of the dynamic range of the variable on each dimension (Eberhart, 2001). In this study, we set initial velocity sets at about 100% of the range of the variables. Inertia weight =0.729 and accelerating factors c1=2.05, c2=2.05 are commonly used parameter values for the PSO (Clerc, M., and J. Kennedy., 2002; Banks et al., 2007, 2008). Considering both the computational time and the optimal precision, we take the population size of the PSO as 60, and the maximum evolutional generation as 200. The results are dependent on the choosing of inertia weight and accelerating factors. Fortunately, 0.729, 2.05 and 2.05 are effective for general optimization problems. In addition, it seems that there is no general set rules of population size and maximum evolutional generation that are suitable for any optimization problems. C3. The authors also mentioned the generic algorithm (GA). If possible, a simple comparison of this approach to the PSO would help the reader a lot to know the advantage of the PSO approach. A3. Thank you very much for your valuable suggestions! We have added content of the comparison in the introduction. Please see Page 3, lines 16-21. C4. P1 Ln10: the CNOP method has been . . . should be the CNOP has been. . . A4. We have deleted the "method". Please see Page 1, line 10. C5. P2 Ln13: Kalnay, 2003 should be Kalnay 2003. A5. We have revised the error. Please see Page 2, line 11. C6. P5 Ln 3: T also depends on u0 . A6. Thank you very much for your valuable suggestions! We have made the corresponding revise. Please see Page 5, line 2. C7. P7 Ln7-8: The Ikeda model is better to rewrite in (x1, x2) instead of (x, y) to make it compatible with note on P8 Ln7. A7. Thank you very much for your helpful suggestions! We have changed the form of Ikeda model in section 2.4. Please see Page 7, lines 7-8. C8. P10L3: are completely same . . . should be are completely the same . . . A8. We have made the corresponding revise. Please see Page 10, line 15.

Please also note the supplement to this comment:
http://www.nonlin-processes-geophys-discuss.net/npg-2016-55/npg-2016-55-AC2-

supplement.zip